# 2D-Vector Magnetic Sensing Based on Ring-Shaped Fiber-Optic Structure Coated with Magnetic Fluid

**DOI:** 10.3390/mi14122140

**Published:** 2023-11-22

**Authors:** Qijing Lin, Liangquan Zhu, Na Zhao, Ping Yang, Zhuangde Jiang

**Affiliations:** 1The Higher Educational Key Laboratory for Flexible Manufacturing Equipment Integration of Fujian Province, Xiamen Institute of Technology, Xiamen 361021, China; xjjingmi@163.com; 2The State Key Laboratory for Mechanical Manufacturing Systems Engineering, Xi’an Jiaotong University, Xi’an 710054, China; zn2020@xjtu.edu.cn (N.Z.); ipe@xjtu.edu.cn (P.Y.); zdjiang@xjtu.edu.cn (Z.J.); 3Shandong Laboratory of Yantai Advanced Materials and Green Manufacturing, Yantai 265503, China; 4Xi’an Jiaotong University (Yantai) Research Institute for Intelligent Sensing Technology and System, Xi’an Jiaotong University, Xi’an 710049, China

**Keywords:** magnetic fluid, magneto-optical characteristic, fiber-optic sensor, 2D vector magnetic sensing

## Abstract

In this work, a novel fiber-optic sensor for 2D magnetic sensing is explored based on nanostructured magnetic fluid. The fiber-optic sensor comprises a ring-shaped fiber structure that is coated with magnetic fluid. The unique magneto-optical characteristic of the nanostructured magnetic fluid enables the fiber-optic structure to detect magnetic fields. By utilizing the 3D Monte Carlo method, the magneto-optical characteristic induced by the nanostructure changes in the magnetic fluid was analyzed. The sensor can realize 2D vector magnetic sensing by intensity demodulation and achieves a sensitivity of 2.402 dB/mT. The proposed fiber optic sensor helps in developing a high-sensitivity 2D vector magnetic field sensor, which has potential applications in the fields of navigation, electrical power systems, and biological detection.

## 1. Introduction

Magnetic field sensing plays a crucial role in such fields as navigation, electrical power systems, and biological detection. Owing to the advantages of low cost, remote detecting, and distributed sensing [1,2,3] the application of fiber-optic sensors provides a promising prospect for magnetic sensing. However, optical fibers do not directly respond to changes in magnetic fields. The emergence of materials such as Terfenol-D [4,5], Al wire [6], Metglas alloy [7], and magnetic fluid (MF) [8] has propelled the advancement of fiber-optic magnetic field sensors. Moreover, the development of oscillators [9] and wireless sensors [10] also holds significant importance in the creation of highly precise quartz sensors, which have been developed, and which take temperature compensation into account.

By exploring MF as a sensing material, fiber-optic magnetic sensors have aroused considerable interest in recent years [11,12,13,14]. Various fiber structures integrated with MF have been developed, including fiber grating [15,16], microfiber [8], microfiber couplers [17], photonic crystal fibers (PCFs) [18,19], Fabry–Perot (FP) interferometers [20], Sagnac interferometers [21], and surface plasmons [11]. Most of the current fiber magnetic field sensors are mainly based on wavelength demodulation methods to demodulate magnetic field information. Generally, changes in light intensity signals are often preferable, given their potential contribution to the miniaturization of sensor systems. Although the light source may cause some deviation in the intensity demodulation, the influence is relatively small and can be solved by a differential method. Thus, many intensity demodulated sensors have been investigated. In 2018, Hu Liang et al. [19] investigated a compact fiber-optic magnetic field sensor by selectively injecting MF into the air core of PCF. MF in PCF can induce strong coupling with light mode transmitted in the optical fiber. In 2019, Zixuan Jia et al. [22] designed a novel temperature self-compensative magnetic field sensor by cascading a single mode fiber (SMF)-no core fiber (NCF)-SMF structure with two fiber Bragg gratings (FBGs). Excited multiple modes in NCF formed the multimode interference, and FBGs were used for temperature compensation. In 2020, Pengfei Li et al. [23] used tapered FBG and MF as bases in proposing an intensity-modulated magnetic field sensor. By tapering FBG, the sensor can generate a powerful evanescent field in the sensing region. In 2021, Yu Tao et al. [24] reported a reflective fiber magnetic field sensor. The fiber sensor is composed of an SMF-NCF-Few core fiber (FCF)-NCF structure and encapsulated by MF. The first section of NCF can lead substantial light into the cladding of FCF and the second NCF is coated with silver as a reflective layer.

As the magnetic field is a vector, it has both intensity and orientation. Most recently, intensity-demodulated fiber magnetic field sensors have mainly focused on magnetic intensity information but ignored orientations, thereby limiting their application. To achieve vector magnetic sensing, in 2018, Weijia Bao et al. [25] inscribed FBG onto multi-clad fiber (MCF) and developed a vector magnetic field sensor. After integrating with MF, the sensitivity of fiber-optic sensors reached 1.43 dB/mT. Similarly, based on MCF, Junying Zhang et al. [26] in 2021 investigated a fiber-optic sensor for vector magnetic sensing and realized 0.353 dB/mT.

In this paper, based on a ring-shaped structure, an intensity demodulation fiber-optic sensor is explored and experimental verified. The structure is simple to fabricate and can excite strong fiber surface evanescent waves on the fiber surface. The MF nanostructure changes under the magnetic field were simulated by the 3D Monte Carlo method to explain the magneto-tropic anisotropy optical property. Combing the ring-shaped fiber structure with MF, the fiber-optic sensor is able to detect both intensity and orientation changes with a high sensitivity by intensity demodulation.

## 2. Materials and Methods

The structure of the ring-shaped fiber is illustrated in Figure 1a, which is fabricated by bending SMF. The stress-optic effects caused by the ring shape induce changes in the refractive index distribution across the fiber cross-section [27]. As shown in Figure 1b, the effective index of guided light mode, denoted as *n_mode_*, is slightly bigger than the cladding *n_clad_* but smaller than core *n_core_*, which is the requirement for light transmission along the optical fiber. However, the ring shape of the fiber will lead to a decrease in the refractive index in the inner part [28,29], while that of the outer part of the bending fiber is increased, as shown in Figure 1c.

As the size of the ring-shaped structure decreases, the *n_mode_* gradually falls below the *n_clad_*. When *n_mode_* falls below the *n_clad_*, the light passing through the ring-shaped region tends to leak into the cladding, generating a fiber-optic surface evanescent field. It is evident that a smaller size induces greater light leakage. To further analyze the effect of the ring shape on light transmission, the ring-shaped optical fiber was equated to a straight optical fiber, and then the Beam Propagation Method (BPM) was used to simulate light transmission, as depicted in Figure 1d. In the simulation, the optical fiber model has a cladding diameter of 125 μm and a core diameter of 8.2 μm. The refractive indices of the fiber cladding and core are 1.4447 and 1.4504, respectively, and the simulated bend radius parameter is set at 4 mm. It can be seen that, owing to changes in the refractive index, input light leaks into the fiber cladding, resulting in a strong fiber-optic evanescent wave propagating along the fiber surface.

After passing the ring-shaped region, light will recouple into the fiber core and interfere with the light guided into the fiber core, which is similar to a Mach–Zehnder interferometer [30]. Consequently, interference light intensity can be approximately written as follows [27,31]:(1)$$I_{out}=I_{co}+I_{cl}+2\sqrt{I_{co}I_{cl}}\cos\left(\frac{2\pi L}{\lambda }\times \Delta n_{eff}\right)$$
where *I_co_* and *I_cl_* represent the intensities of light propagated in the fiber core and cladding, *L* stands for the inference length, and Δ*n_eff_* represents the effective refractive index difference for the light guided into the fiber core and cladding.

MF, known as one kind of functional material, is composed of 10 nm magnetic nanoparticles dispersed in the fluid. The fluid has both the magnetic properties of magnetic materials and the liquidity of liquids, which results in many interesting magneto-optic properties. These unique physical optical characteristics are closely related to its nanostructures. In MF, the behavior and properties of the nanoparticles are influenced by various micro-interactions. These interactions play a crucial role in determining the overall behavior of the MF. The primary interactions involve a van der Waals interaction, surfactant repulsive interaction, magnetic dipole–dipole interaction, and magnetic dipole–field interaction [32]. The van der Waals potential energy is mathematically expressed as:(2)$$E_{VDW}=-\frac{A}{6}\left[\frac{2}{L_{V}^{2}+4L_{V}}+\frac{2}{{\left(L_{V}+4\right)}^{2}}+\ln\frac{L_{V}^{2}+4L_{V}}{{\left(L_{V}+4\right)}^{2}}\right]$$
(3)$$L_{V}=\frac{r_{ij}}{d}-2$$

Here, *A* represents the Hamaker constant, *r_ij_* signifies the distance between particles *i* and *j*, and *d* denotes the diameter of the particles.

And, the surfactant repulsive interaction between particles can be formulated as:(4)$$E_{STERIC}=\frac{\pi d^{2}\zeta kT}{2}\left(2-\frac{r_{ij}}{\delta }\ln\left(\frac{\delta }{r_{ij}}\right)-\frac{r_{ij}-d}{\delta }\right)$$

Here, *ζ* stands for the surface density of surfactant molecules and *δ* denotes the thickness of the surfactant molecules.

The potential energy associated with the dipole–dipole interaction between particles can be defined as:(5)$$E_{DIPOLE}=\frac{\mu_{0}m}{4\pi d^{3}}\left(\frac{d_{p}}{r_{ij}}\right)\left[\left(n_{i}\cdot n_{j}\right)-3\left(n_{i}-t_{ij}\right)\left(n_{i}-t_{ij}\right)\right]$$

Here, *μ*_0_ represents the magnetic permeability of free space; *m* signifies the magnetic moment of ferromagnetic particles; *t_ij_* denotes the unit position vector between particles *i* and *j*; while *n_i_* and *n_j_* represent the magnetic moment directions of *i* and *j,* respectively.

When an external magnetic field is applied, the energies associated with magnetic dipole–field interactions are expressed as:(6)$$E_{MAGNETIC}=\mu_{0}mH$$

Here, *H* represents the external magnetic field intensity.

To better understand the nanostructure changes, a 3D Monte Carlo program based on Equations (2)–(6) was developed to simulate in MATLAB 2020b. As shown in Figure 2a, without an externally applied magnetic field, these nanoparticles are distributed in a disorderly manner. The MF presents isotropic optical properties, i.e., as light in different directions passes the MF, the transmission loss of light is the same. Thus, the whole MF can be treated as an isotropic material. In addition, because of the disorderly distribution of these nanoparticles, the magnetic torque between particles can offset itself, so the overall MF shows no magnetism. However, as illustrated in Figure 2b,c, with the emergence of the magnetic torque, the MF presents a different optical property, i.e., magnetic nanoparticle chains are formed along the magnetic field direction, which will change the light absorption, scattering coefficients, and refractive index [32,33,34,35]. The appearance of nanoparticle chains changes the magnetic fluid into an anisotropic material, i.e., the transmission loss of light in different directions is different. When the light propagates along the magnetic field, the transmission loss of light is decreased. While light propagates perpendicular to the magnetic field, the transmission loss of light is increased. Based on the optical property difference in the vertical and parallel magnetic field directions, the 2D Vector Magnetic sensing can be achieved by combining the fiber-optic sensors and the MF.

## 3. Results and Discussion

To verify the performance of the fiber sensor, the experiment setup was built as depicted in Figure 3a, which includes a magnetic field generator, DC power supply, broad light source, and a spectrometer. To generate a 2D magnetic field, the generator is fixed on a rotatable stage that can be rotated from 0° to 360°. In the calibration tests, a Tesla meter with a resolution of 0.01 mT was used to monitor the real-time magnetic field changes. The ring-shaped fiber sensor was placed in a cylindrical container, which was filled with MF (EMG605) as illustrated in Figure 3b. In addition, for simplification of the analysis, the orientation of the magnetic field vertical to the fiber-optic sensor was defined as 0°/180°, while the parallel direction was defined as 90°/270°.

### 3.1. Response to Dual-Directions Magnetic Field Intensity

As discussed before, the ring structure will excite the fiber evanescent wave propagating along the fiber surface. With the decrease in ring shape size, more light will be excited from the fiber core to the evanescent wave, which can enhance the sensitivity. As the MF will change into an anisotropic material under external magnetic field, by inserting the fiber structure into the MF, the magnetic field can be detected by monitoring the output light transmission loss.

From Equation (1), the maximal intensity will appear when the accumulated optical phase equals 2*n*π (*n* is an integer). And thus, the peak intensity can be expressed as:(7)$$I_{pe}={\left(\sqrt{I_{co}}+\sqrt{I_{cl}}\right)}^{2}$$

Owing to these magnetic nanoparticles increasing the absorption and scattering coefficients, spectrum intensity will be decreased after the fiber probe is inserted into the MF. Peak intensity of output light can be written as follows [32]:(8)$$I_{po}^{M}=I_{po}\exp\left(-\alpha rL\right)$$
where *α* is the absorption coefficients of MF and *r* represents the ratio of the fiber-optic evanescent field intensity to all propagated.

Thus, the peak intensity changes can be adopted to demonstrate the external magnetic field changes. As shown in Figure 4, after the fiber structure was placed instead into the MF with air as the medium, the peak intensity output spectrum intensity is decreased, which is due to the optical properties difference between air and MF, and part of the light was absorbed by MF.

Then, to verify the sensor’s performance, the response to the dual-direction (0° and 90°) magnetic field was investigated. Figure 5 shows the variation of the output spectrum of sensors in dual directions, respectively. The peak intensity decreases with the magnetic field intensity increasing at 0°, while those at 90° are slightly increasing. As depicted in Figure 5c,f,i, by linear fitting, the sensors with different ring-shaped structures show sensitivities of 0.550, 1.361, and 2.402 dB/mT at 0°, and 0.009, 0.027, and 0.048 dB/mT at a 90° direction, respectively. Apparently, the smaller ring-shaped structure size has a higher sensitivity, which is because the smaller ring-shaped structure excites the stronger evanescent wave. It is worth noting that at 0°, the intensity does not change linearly. This behavior occurs because the MF initially magnetizes slowly and eventually saturates at higher magnetic field intensities.

The difference in the sensor’s response to the dual-direction magnetic field can be explained by Figure 6. At 0°, the formed chain structures in MF are vertically distributed relative to the fiber structure. As the fiber optic evanescent wave propagates along the fiber surface, these chain structures act as obstacles, blocking the light and increasing the absorption and scattering coefficients within the magnetic fluid, while at 90°, the chains in the magnetic fluid align parallel to the fiber structure, which are also parallel to the direction of the fiber optic evanescent wave propagation. As a result, the absorption and scattering coefficients of the magnetic fluid are slightly reduced compared to the 0° angle case. The response difference observed in the sensor’s output for different magnetic field angles can be attributed to the arrangement of chain structures in the magnetic fluid. These nanostructures distributed in MF affect the absorption and scattering coefficients of the magnetic fluid and influence the sensor’s response.

### 3.2. Response to Magnetic Field Directions

Due to the nanostructures in MF shifting with the external magnetic field, the fiber-optic sensor’s response to dual directions is opposite. To further characterize the response to the continual direction changes in the 2D magnetic field, the intensity was maintained at 9 mT, and then the magnetic field generator was rotated. With the rotating of the generator, the magnetic orientation changed from 0° to 360°. The output spectrum was recorded every 10° to show the difference. Figure 7a–d shows spectrum changes with different directions. The peak intensity decreases in the ranges of 0–90° and 180–270°, which correspond to the chain structure’s direction changing from a parallel direction to a vertical direction. And in the range of 90–180° and 270–360°, the peak intensity increases. To better understand the peak intensity changes in the whole 2D magnetic field, the resonance dip wavelength variations in the polar and Cartesian coordinate systems are plotted in Figure 7e and Figure 7f, respectively. However, the patterns are not perfectly symmetrically in the 2D magnetic field, which may be caused by the fiber-optic sensor not being fabricated symmetrically. As the sensor’s response to different magnetic directions are different, the fiber sensor has the potential to realize the 2D vector magnetic sensing by intensity demodulation. To determine the magnetic field’s direction, a rotation of the fiber-optic sensor is necessary to identify the positions of two minimum peak intensities (just working like a Rader) [36]. These positions align parallel to the orientation of the magnetic field. It also should be pointed out that the sensor’s orientation detection is subject to an uncertainty of 180°, meaning it cannot discern the sign of the magnetic field direction [36,37,38]. Once this is done, the magnitude of the peak intensity establishes a direct correlation with the magnetic field direction, providing a one-to-one relationship.

To assess the directional accuracy of the ring-shaped fiber-optic sensor, data spanning from 0° to 180° were extracted, as depicted in Figure 8, and data of the direction ranging from 0° to 180° were extracted, as shown in Figure 8. Peak intensity changes are not linear with the magnetic direction over the entire range. However, there are linear relationships in the ranges of 10–50°, 50–90°, 90–130°, and 130–170°. By applying linear fitting, the magnetic field direction sensitivities of 0.210 nm/° and 0.080 nm/° were reached in the direction ranges of 10–50°and 50–90°, respectively. The corresponding measurement errors are 1.4% and 0.6% respectively. In the 90–130° and 130–170° ranges, magnetic field direction sensitivities are 0.089 nm/° and 0.201 nm/°, respectively. And the corresponding measurement errors are 0.7% and 3.2%, respectively.

### 3.3. Response to Temperature

To analyze the temperature cross-sensitivity, another fiber-optic magnetic field sensor with a short axis of 6 mm was fabricated. Figure 9a,b depicts the output spectrum variation of the sensor at directions 0° and 90°, respectively. In the 0° direction, the peak intensities of the output spectrum decrease with the rise in magnetic field intensities, while at the 90° direction, the peak intensities exhibit a slight increase, consistent with other fiber magnetic field sensors. As illustrated in Figure 9c, the sensitivity reaches 3.92 dB/mT in the range of 4–8 mT, while at the direction of 0°, the fiber sensor has a sensitivity of 0.23 dB/mT.

Then, the fiber-optic magnetic field sensor was placed in a vacuum drying oven (DZF-6024, Shanghai bluepard instruments Co., Ltd., Shanghai, China). In the vacuum drying oven, no vacuum was applied, and the pressure inside was consistent with the ambient pressure. The recorded transmission spectra from 30 °C to 50 °C are presented in Figure 10a. The peak intensity gradually increases with temperature variations. Figure 10b illustrates that the peak intensity changes are relatively small throughout the temperature measurement range. Employing a linear regression to the peak intensity across various temperatures, a temperature sensitivity of 0.04 dB/°C is attained within the 30 °C to 50 °C span. It is worth noting that compensating for the crosstalk between temperature and magnetic field can be achieved by monitoring the operational temperature [18,22,39].

### 3.4. Comparison and Discussion

The performance of the fiber-optic sensor was compared to other intensity-demodulated magnetic sensors in recent years, as shown in Table 1. The proposed sensor exhibits higher sensitivity compared to the others. Additionally, the sensor offers the advantage of a simple fabrication method when compared to alternative sensors. The unique fiber-optic structure enables the sensor to detect both intensity and orientation changes with a high sensitivity by intensity demodulation.

## 4. Conclusions

In summary, a novel intensity-demodulated fiber-optic sensor that functionalizes with MF is presented. The sensor was fabricated using a simple method just by bending SMF. By using the 3D Monte Carlo method, the magneto-tropic property of MF was explained. By taking advantage of the strong evanescent field generated by the structure and the magneto-tropic property of MF, the peak intensities of the output spectrum were found to change in response to the intensity and orientation of an external magnetic field. The intensity and orientation information can be demodulated from the changes in the output spectrum. The sensor exhibited a high sensitivity of 2.402 dB/mT, indicating its potential for various applications.

## Figures and Tables

**Figure 1 micromachines-14-02140-f001:**
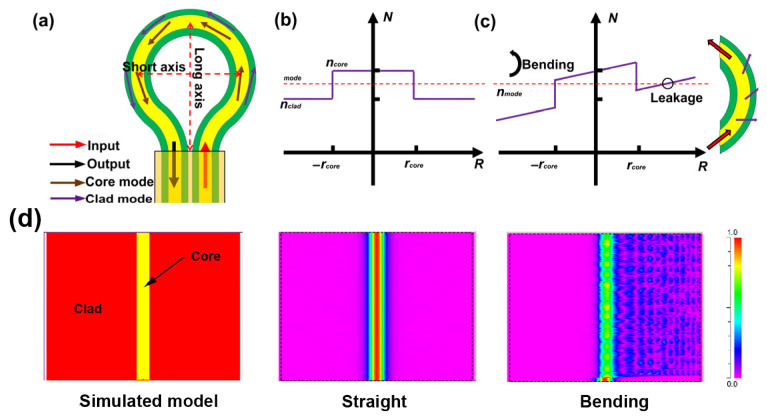
(**a**) The ring shape-sensing structure. Refractive index of (**b**) straight optical fiber; (**c**) bending optical fiber. (**d**) Light distribution simulated in straight and bending optical fibers.

**Figure 2 micromachines-14-02140-f002:**
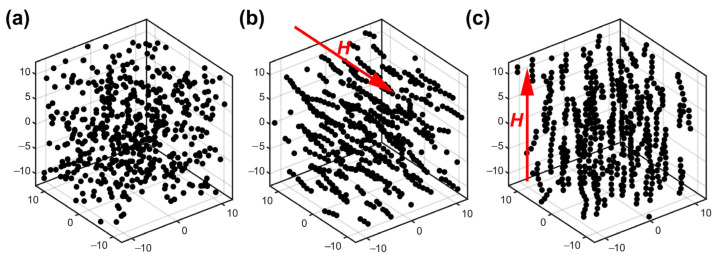
The 3D Monte Carlo simulation of magnetic nanoparticles (**a**) without external applied magnetic field; (**b**) in the parallel magnetic field (5 mT); (**c**) in the vertical magnetic field (5 mT).

**Figure 3 micromachines-14-02140-f003:**
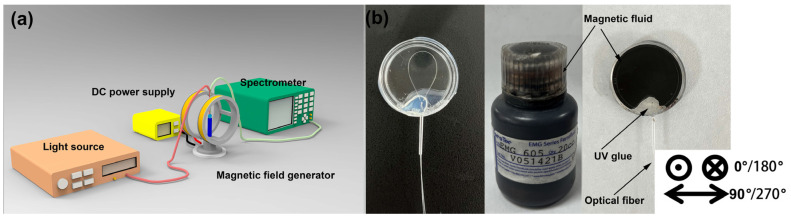
Schematic of the (**a**) experiment setup and (**b**) fiber-optic magnetic sensors.

**Figure 4 micromachines-14-02140-f004:**
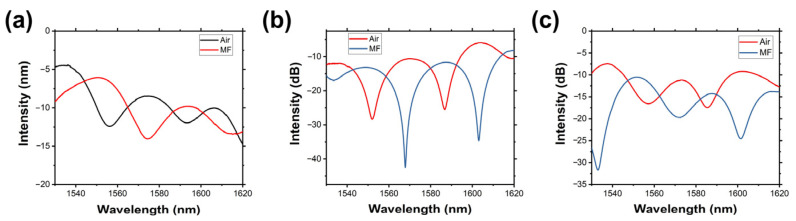
The output spectrum of the ring-shaped structure in air and MF of the short axis of (**a**) 12 mm, (**b**) 10 mm, and (**c**) 8 mm.

**Figure 5 micromachines-14-02140-f005:**
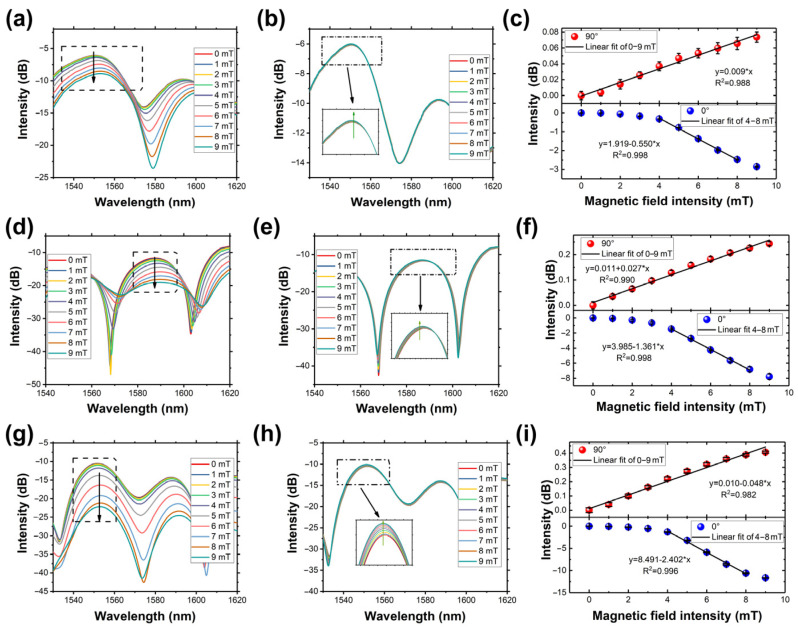
Spectrum response of 12 mm short axis ring-shaped fiber structure to the external magnetic field at the directions of (**a**) 0° and (**b**) 90°. (**c**) Peak intensity varies in dual directions. Spectrum response of 10 mm short axis ring-shaped fiber structure to the external magnetic field at the directions of (**d**) 0° and (**e**) 90°. (**f**) Peak intensity varies in dual directions. Spectrum response of 8 mm short axis ring-shaped fiber structure to the external magnetic field at the directions of (**g**) 0° and (**h**) 90°. (**i**) Peak intensity varies in dual directions.

**Figure 6 micromachines-14-02140-f006:**
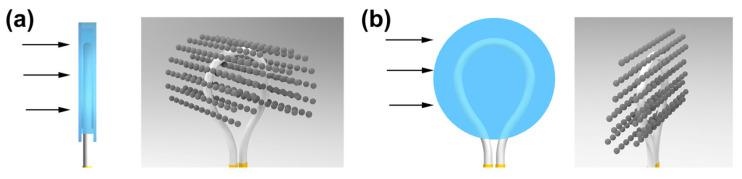
Schematic diagram of the nanostructures in MF with the fiber-optic structure at the directions of (**a**) 0° and (**b**) 90°.

**Figure 7 micromachines-14-02140-f007:**
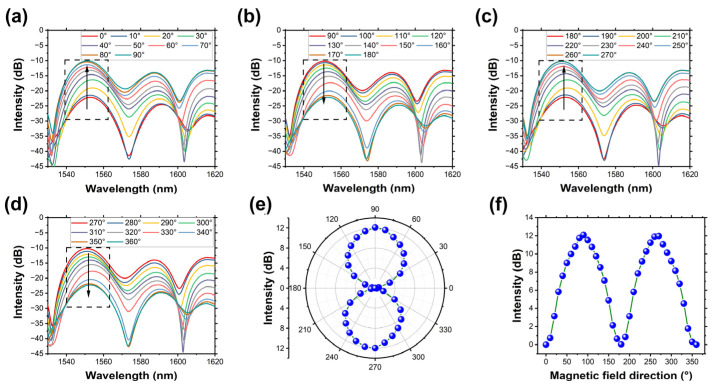
(**a**–**d**): The output spectrum in the magnetic direction range of 0–360°. The variations of peak intensities plotted in the (**e**) polar coordinate system, and (**f**) Cartesian coordinate system.

**Figure 8 micromachines-14-02140-f008:**
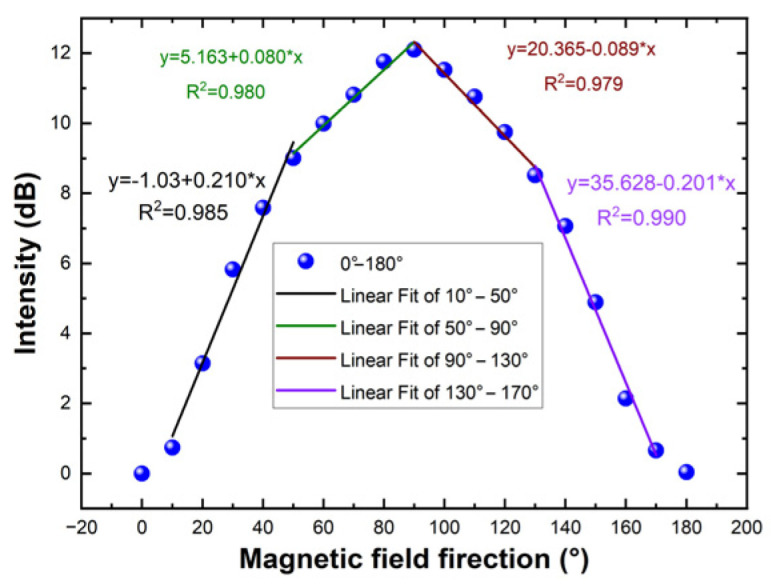
The peak intensity varies with magnetic field direction in 0–180°.

**Figure 9 micromachines-14-02140-f009:**
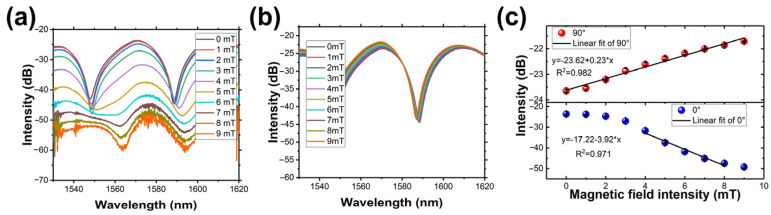
The spectrum response of the 6 mm short axis ring-shaped fiber structure to the external magnetic field in the directions of (**a**) 0° and (**b**) 90°. (**c**) The peak intensity varies with external magnetic field in two directions.

**Figure 10 micromachines-14-02140-f010:**
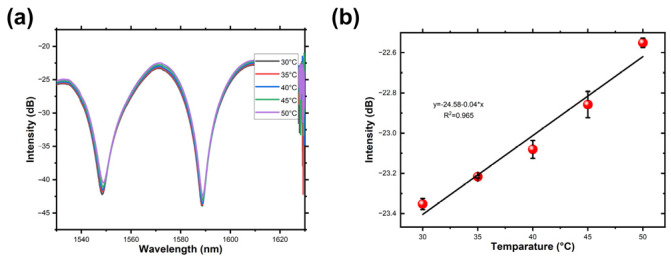
(**a**) The transmission spectra within the temperature range of 25–50 °C. (**b**) The peak intensity varies with temperature.

**Table 1 micromachines-14-02140-t001:** Performance comparison of fiber-optic sensors for vector magnetic sensing.

Structures	Fabrication Method	Vector	Sensitivity	Ref.
Dual S-tapered multimode fiber	Spicing and tapering fibers	No	0.11 dB/mT	[35]
SMF-PCF-SMF	Fill the air core of PCF with MF	No	0.19 dB/mT	[19]
SMF-no core fiber (NCF)-SMF	Spicing NCF between two SMF	No	1.28 dB/mT	[22]
Tapered fiber Bragg grating	Using the heating-pull method with *H*_2_ flame	No	1.933 dB/mT	[23]
SMF-NCF-FCF-NCF structure	Spicing fibers	No	1.20993 dB/mT	[24]
FBG	Inscribe FBG on fiber	Yes	1.43 dB/mT	[25]
FBG	Inscribe FBG on fiber	Yes	0.353 dB/mT	[26]
Ring-shaped structure	Bending fiber	Yes	2.402 dB/mT	-

## Data Availability

The original contributions presented in the study are included in the article. Further inquiries can be directed to the corresponding author.

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
