# Peer review of "2D-Vector Magnetic Sensing Based on Ring-Shaped Fiber-Optic Structure Coated with Magnetic Fluid"

_micromachines, 2023, doi:10.3390/mi14122140_

Round 1
Reviewer 1 Report
Comments and Suggestions for Authors
A magnetic field sensor is proposed and demonstrated in this manuscript. The sensor is made up of a droplet fiber ring that is immersed in magnetic fluid and is sensitive to the intensity and orientation of the magnetic field. The authors claim that it can be used as a 2D-vector magnetic field sensor, however I'm not sure how to tell whether the intensity variation is caused by magnetic field intensity or direction; this should be clarified. Otherwise, this function will be a disadvantage rather than a benefit of the sensor. It is advised that the 3D Monte Carlo simulation be introduced more explicitly. What about the influence of the ambient temperature? and how to deal with it. I cannot provide a positive recommendation unless the following issues are addressed clearly.
1. The authors claim that “Owing to the advantages of low cost, remote detecting, distributed sensing, and anti-electromagnetic interference [1–3], the application of fiber-optic sensors provides a promising prospect for magnetic sensing.” How to understand the advantage of “anti-electromagnetic interference” for a magnetic field sensor?
2. In section of Introduction, “By integrating MF with optical fiber, many photonic devices have been developed, such as optical switches [9], tunable filters [10], magneto-optical modulators 34 [11], and magnetic field sensors [12-15].” The Ref. [9-11] are not closely related to the topic of this manuscript.
3. In section of Introduction, “However, in the field of fiber-optic sensors, wavelength changes of light signals are relatively difficult to detect and demodulate,” it is not accurate to say so. The authors state in this manuscript that the proposed sensor operates on intensity demodulation rather than wavelength demodulation, but how do they trace a peak and calculate its intensity when the magnetic field changes? After all, the resonance peak shifts with the variation of surrounding magnetic field. Do you believe that this so-called intensity-demodulation is simpler than wavelength demodulation?
4. What does “ans” mean in the sentence of “the influence is relatively ans can be solved by differential method”?
5. Please provide literatures to support this opinion of “the fiber will lead to the decrease of refractive index in the inner part”.
6. In Fig. 1, please provide the model parameters for simulate bending fiber.
7. Since the fiber ring looks like a droplet rather than a circle, it is suggested to describe it by using the long and short axes, rather than using the diameter.
8. The experimental results reveal that the sensor is sensitive to both magnetic field intensity and orientation, but how do you differentiate the light intensity fluctuation caused by magnetic field intensity or orientation? How does 2D-Vector Magnetic Sensing work?
9. 3D Monte Carlo simulation is suggested to introduced more clearly.
10. What about the influence of the ambient temperature? and how to deal with it.
Comments on the Quality of English LanguageA magnetic field sensor is proposed and demonstrated in this manuscript. The sensor is made up of a droplet fiber ring that is immersed in magnetic fluid and is sensitive to the intensity and orientation of the magnetic field. The authors claim that it can be used as a 2D-vector magnetic field sensor, however I'm not sure how to tell whether the intensity variation is caused by magnetic field intensity or direction; this should be clarified. Otherwise, this function will be a disadvantage rather than a benefit of the sensor. It is advised that the 3D Monte Carlo simulation be introduced more explicitly. What about the influence of the ambient temperature? and how to deal with it. I cannot provide a positive recommendation unless the following issues are addressed clearly.
1. The authors claim that “Owing to the advantages of low cost, remote detecting, distributed sensing, and anti-electromagnetic interference [1–3], the application of fiber-optic sensors provides a promising prospect for magnetic sensing.” How to understand the advantage of “anti-electromagnetic interference” for a magnetic field sensor?
2. In section of Introduction, “By integrating MF with optical fiber, many photonic devices have been developed, such as optical switches [9], tunable filters [10], magneto-optical modulators 34 [11], and magnetic field sensors [12-15].” The Ref. [9-11] are not closely related to the topic of this manuscript.
3. In section of Introduction, “However, in the field of fiber-optic sensors, wavelength changes of light signals are relatively difficult to detect and demodulate,” it is not accurate to say so. The authors state in this manuscript that the proposed sensor operates on intensity demodulation rather than wavelength demodulation, but how do they trace a peak and calculate its intensity when the magnetic field changes? After all, the resonance peak shifts with the variation of surrounding magnetic field. Do you believe that this so-called intensity-demodulation is simpler than wavelength demodulation?
4. What does “ans” mean in the sentence of “the influence is relatively ans can be solved by differential method”?
5. Please provide literatures to support this opinion of “the fiber will lead to the decrease of refractive index in the inner part”.
6. In Fig. 1, please provide the model parameters for simulate bending fiber.
7. Since the fiber ring looks like a droplet rather than a circle, it is suggested to describe it by using the long and short axes, rather than using the diameter.
8. The experimental results reveal that the sensor is sensitive to both magnetic field intensity and orientation, but how do you differentiate the light intensity fluctuation caused by magnetic field intensity or orientation? How does 2D-Vector Magnetic Sensing work?
9. 3D Monte Carlo simulation is suggested to introduced more clearly.
10. What about the influence of the ambient temperature? and how to deal with it.
Reviewer 2 Report
Comments and Suggestions for Authors
In this paper, a magnetic sensor based on magnetic fluid (MF) and ring-shape fiber is investigated for 2D-Vector magnetic sensing. 3D Monte Carlo method was used to explain the magneto-tropic property of MF in theory. Experimentally, the 2D-Vector magnetic sensing was demonstrated and a competitive sensitivity was achieved. It’s a topic of interest to researcher in the related areas. The paper is well-written and can be published with minor revisions. My detailed comments are as follows:
1). There are several typo errors in this paper, such as “ans” (Line 47), “c”(Line 57), and “ncore” (Line 82). Authors should pay close attention to these issue and make the necessary corrections.
2). From figure 5, the light intensity response to magnetic field intensity shows the different trends at 0o and 90o. Especially, the light intensity change depending magnetic field is much slighter within 0-4mT for 0o case. Authors should add explanations on the above phenomena.
3). The magnitude of the magnetic field used in Monte Carlo simulations should be clearly marked in the paper (see Figure. 2).
4). In line 107, I recommend replacing 'intelligent materials' with 'functional materials'.
5). Please check the page numbers on each page of the paper to ensure they are correctly ordered and correspond to the actual page numbers.
Comments on the Quality of English LanguageEnglish is fine. Minor suggestions have been included in the above comments.
Reviewer 3 Report
Comments and Suggestions for Authors
Page 1: line 35: Athors should add ref. below and add: ”This research is also very important for new very precise quartz sensors which have been developed, and which take temperature compensation into account:”
-Detection principles of temperature compensated oscillators with reactance influence on piezoelectric resonator. Sensors. 2020, vol. 20, iss. 3, p. 1-18. ISSN 1424-8220. https://www.mdpi.com/1424-8220/20/3/802
-Yang, S.; Tan, M.; Yu, T.; Li, X.; Wang, X.; Zhang, J. Hybrid Reduced Graphene Oxide with Special Magnetoresistance for Wireless Magnetic Field Sensor. Nano-Micro Letters 2020, 12, 1-14, doi:10.1007/s40820-020-0403-9.
Page 5: Figure 7.: How accurately would it be possible to read directions in degrees?
Page 5: How accurately would it be possible to read directions in degrees? Is there any effect of temperature on the reading or detection of the direction?
General: What is the effect of nearby ferrous objects on sensor direction detection in your method from the sensitivity shown?
Round 2
Reviewer 1 Report
Comments and Suggestions for Authors
Except for temperature compensation, i.e., the last issue, other issues are basically resolved in the revision.
I am puzzled why the author particularly emphasized placing the sensor in a vacuum drying oven. When it is used for magnetic field sensing, it should be called a magnetic field sensor together with the magnetic fluid, so the temperature response I need is the temperature response of the fiber optic device immersed in the magnetic fluid.
Please confirm the temperature response of the sensor with magnetic fluid coating.
Comments on the Quality of English LanguageFine, it can be improved.
Round 3
Reviewer 1 Report
Comments and Suggestions for Authors
I accept the revision.
Comments on the Quality of English LanguageFine, it can be improved.